# Real-Life Diagnostic Accuracy and Clinical Utility of Hepatitis B Virus (HBV) Nucleic Acid Testing Using the GeneXpert Point-of-Care Test System from Fresh Plasma and Dry Blood Spot Samples in The Gambia

**DOI:** 10.3390/microorganisms12112273

**Published:** 2024-11-09

**Authors:** Amie Ceesay, Sainabou Drammeh, Gibril Ndow, Alpha Omar A. Jallow, Haddy Nyang, Baboucarr Bittaye, Francis S. Mendy, Ousman Secka, Umberto D’Alessandro, Yusuke Shimakawa, Erwan Vo-Quang, Barbara Testoni, Mark Thursz, Maud Lemoine, Isabelle Chemin

**Affiliations:** 1Disease Control and Elimination, Medical Research Council Unit, The Gambia at the London School of Hygiene and Tropical Medicine, Fajara, Banjul P.O. Box 273, The Gambia or amceesay@utg.edu.gm (A.C.); saindrammeh@mrc.gm (S.D.); gibril.ndow@lshtm.ac.uk (G.N.); aojallow@outlook.com (A.O.A.J.); hnyang@mrc.gm (H.N.); babittaye@mrc.gm (B.B.); fmendy@mrc.gm (F.S.M.); ousman.secka@lshtm.ac.uk (O.S.); umberto.dalessandro@lshtm.ac.uk (U.D.); erwan.voquang@gmail.com (E.V.-Q.); 2Cancer Research Center of Lyon, INSERM U1052, CNRS UMR 5286, 69008 Lyon, France; barbara.testoni@inserm.fr; 3School of Arts and Sciences, University of The Gambia, Serrekunda, Banjul P.O. Box 3530, The Gambia; 4Unité d’Épidémiologie des Maladies Émergentes, Institut Pasteur, Université Paris Cité, 75015 Paris, France; yusuke.shimakawa@pasteur.fr; 5Institut Mondor de Recherche Biomédicale (INSERM U955), 94000 Creteil, France; 6Division of Digestive Diseases, Section of Hepatology, Department of Metabolism, Digestion and Reproduction, Imperial College London, London W2 1NY, UK; m.thursz@imperial.ac.uk

**Keywords:** monitoring, GeneXpert, diagnosis, resource limited, scale-up, settings, utility, HBV, clinical

## Abstract

The GeneXpert HBV Viral Load test is a simplified tool to scale up screening and HBV monitoring in resource-limited settings, where HBV is endemic and where molecular techniques to quantify HBV DNA are expensive and scarce. However, the accuracy of field diagnostics compared to gold standard assays in HBV-endemic African countries has not been well understood. We aim to validate the diagnostic performance of the GeneXpert HBV Viral Load test in freshly collected and stored plasma and dried blood spot (DBS) samples to assess turn-around-time (TAT) for sample processing and treatment initiation, to map GeneXpert machines and to determine limitations to its use in The Gambia. Freshly collected paired plasma and DBS samples (n = 56) were analyzed by the GeneXpert test. Similarly, stored plasma and DBS samples (n = 306, n = 91) were analyzed using the GeneXpert HBV test, in-house qPCR and COBAS TaqMan Roche. The correlation between freshly collected plasma and DBS is r = 0.88 with a mean bias of −1.4. The GeneXpert HBV test had the highest quantifiable HBV DNA viremia of 81.4% (n = 249/306), and the lowest was detected by in-house qPCR at 37.9% (n = 116/306) for stored plasma samples. Bland–Altman plots show strong correlation between GeneXpert and COBAS TaqMan and between GeneXpert and in-house qPCR with a mean bias of +0.316 and −1.173 log_10_ IU/mL, respectively. However, paired stored plasma and DBS samples had a lower mean bias of 1.831 log_10_ IU/mL, which is almost significant (95% limits of agreement: 0.66–3.001). Patients (n = 3) were enrolled in the study within a TAT of 6 days. The GeneXpert HBV test displayed excellent diagnostic accuracy by detecting HBV viremia in less than 10 IU/mL.

## 1. Introduction

Hepatitis B virus (HBV) infection is a major global health issue. The World Health Organization (WHO) estimates that 254 million people are affected by chronic hepatitis B (CHB), with 1.5 million new infections each year, mainly observed in Africa [1]. Globally, mortality due to HBV-related diseases ranges from 500,000 to 1.2 million cases annually, 100,000 of which are attributable to hepatocellular carcinoma (HCC) alone in Africa [2]. The prevalence of the hepatitis B virus surface antigen (HBsAg) is the highest in the WHO African (7.5%) and Western Pacific (6.2%) regions, together representing 68% of the global disease burden (175 million). This burden is increasing in Africa and Asia, with levels as high as 8% in the general adult population, and remains a major cause of morbidity and mortality [3,4,5]. The WHO has developed a strategy to eliminate viral hepatitis that includes improving its diagnosis among others, with the aim of reducing new infections by 90% and deaths by 65% by 2030 [6,7]. It is essential to scale up screening, assess treatment eligibility and provide easily deployable HBV DNA quantification assays in order to achieve this goal.

HBV nucleic acid testing is costly and not easily accessible to those living in low- and middle-income countries (LMICs), where CHB is endemic [8,9]. In Africa, current screening tools rely mainly on serology, rapid point-of-care tests and real-time polymerase chain reaction (RT-PCR). The latter is a molecular assay used to quantify HBV genomic DNA and is often restricted to well-equipped laboratories that are often based in large cities or capitals. In this context, PCR assay platforms require dedicated infrastructure and highly trained laboratory technicians, which often results in a sample turn-around time (TAT) of days or even weeks and high testing costs for patients [10]. Thus, in its 2024 revised guidelines for the management of hepatitis B [7], the WHO has emphasized the urgent need for simplified virological testing, including assays that use point-of-care (POC) nucleic acid technologies (NATs) with ensured affordability and scalability for the quantification of HBV DNA [7].

In addition, in LMICs, access to routine and effective HBV molecular diagnostics tools in order to determine treatment eligibility and monitor treatment response is relatively low. Where available, HBV DNA measurement is expensive, with some commercial assays costing between 73 and 150 €/test [1,11]. Consequently, decentralizing and scaling up HBV treatment programs remains a major challenge in almost all HBV-endemic countries. It is, therefore, urgent to find reliable, cost-effective alternatives to HBV molecular diagnosis for Africa, a region that has been identified as a priority for HBV research due to the high disease burden and absence of guidelines adapted to the local setting [12].

The new generation of HBV DNA real-time assay platforms is cheap compared to other HBV DNA tests and can provide results in less than 2 h and continuous loading of specimens with true random access. The cost for conventional hepatitis B nucleic acid testing is estimated to range from US$30 to 120; for low- and middle-income countries, the cost can go up to US$400 per test [13]. The Foundation for Innovative New Diagnostics (FIND) has negotiated the price of the GeneXpert HBV viral load assay for 145 developing countries, and it costs US$14.9 per cartridge, excluding shipment [14]. The automated GeneXpert rapid molecular system (Cepheid AB, Röntgenvägen 5, SE-171 54 Solna, Sweden) is a commercially available HBV DNA assay approved by the American Food and Drug Association (FDA) and the Therapeutic Goods Administration (TGA) in Australia [14,15,16,17]. It is based on a cartridge in which nucleic acid extraction, amplification and detection of target sequences are automatically carried out. The system has been used for several other pathogens, including hepatitis C virus (HCV) [18,19], human immunodeficiency virus 1 (HIV-1) [20], Mycobacterium Tuberculosis (MTB) [21,22,23] and COVID-19 [24]. The GeneXpert technology revolutionized the management of MTB in LMICs, where it has been used with great success in test-and-treat strategies against this pathogen [25,26,27]. As a result, GeneXpert machines have become ubiquitous in HBV-endemic regions of sub-Saharan Africa (sSA) and Asia.

Thus, the newly launched GeneXpert HBV Viral Load cartridge, designed to detect and quantify HBV DNA by targeting the preC-C (Pre-Core-Core region), could significantly improve and scale up HBV diagnosis and monitoring in LMICs. However, the diagnostic accuracy of this platform in HBV-endemic regions in sSA has not been determined previously. Therefore, the Prevention of Liver Fibrosis and Cancer in Africa (PROLIFICA) project, the first HBV “screen and treat” program in West Africa, provides an excellent opportunity to fill this knowledge gap. PROLIFICA follows a cohort of 1200 adults in The Gambia. Of these, 150 receive antiviral therapy, and the remaining 1050 are at various stages of HBV infection and levels of viremia [3]. Using samples from these participants, our aim was to (a) compare the diagnostic accuracy of the Xpert HBV Viral Load test in quantifying HBV viremia against the gold standard (PCR assays) in an African setting using three types of samples, namely stored plasma, freshly collected plasma and dried blood spot (DBS) samples, (b) determine the TAT for sample processing and treatment initiation and (c) map GeneXpert machine use and real-world challenges faced by healthcare facilities using the GeneXpert assay in The Gambia.

## 2. Materials and Methods

### 2.1. Study Participants

This study is a cross-sectional analysis nested within the PROLIFICA project, which was conducted between 7 December 2011 and 24 January 2014. The PROLIFICA program was a large-scale cohort of chronic hepatitis B HBsAg-positives established from a community-based and blood bank-based screening. At baseline, a finger-prick whole-blood test for HBsAg was carried out using a rapid point-of-care (POC) test (Determine, Alere, Waltham, MA, USA), with a field performance of sensitivity and specificity of 99.0 (94.5–100.0) and 99.4 (96.9–100.0 at 95 CL), respectively [28]. All participants had socio-demographic questionnaires, anthropometric (weight, height) and medical history data collection, physical examination and comprehensive liver assessment, including a fasting Fibroscan (Echosens, Paris, France), abdominal ultrasound and liver biopsy, when indicated. Patients eligible for antiviral treatment according to the European Association for Study of the Liver (EASL) guidelines were offered tenofovir disoproxil fumarate (TDF), 300 mg daily [3,29,30], free of charge. The study was approved by The Gambia Government/Medical Research Council The Gambia Unit (MRCG) Joint Ethics Committee (SCC1266-V2).

During enrollment, contact details (phone numbers, home address and two trusted relatives) were recorded. Between 2013 and 2021, participants were contacted regularly by the study team to ascertain their survival status. In October 2018, all CHB participants were invited to a comprehensive reassessment. Following an informed consent, participants completed socio-demographic and clinical questionnaires and underwent a physical examination as performed at baseline. This phase of the PROLIFICA study was approved by The Gambia Government/Medical Research Council The Gambia Unit (MRCG) Joint Ethics Committee (SCC1579).

#### 2.1.1. Sample Collection

Two groups of treatment-naïve patients were analyzed for the validation study. For group 1, paired plasma and DBS samples freshly collected from 56 HBsAg-positive patients (unknown viral loads) were analyzed within two months (March 2023 to April 2023). Venous blood (4 mL) was collected in ethylenediaminetetraacetic acid (EDTA) or serum separator tubes (SSTs) from each participant. Plasma and serum were separated within 4 h of collection. DBS samples were prepared at the same time as venous blood collection using 50 µL/spot on two spots on Whatman 903 Protein Saver cards™ (GE Healthcare Europe, Freiburg, Germany), air dried for 2 h and stored in zip-lock bags with desiccant until further use.

The second group included paired plasma and DBS samples, which were collected over three months one year prior to this study (November 2019 to January 2021). This group is referred to as stored samples. Here, venous blood of plasma samples from 306 HBsAg-positive patients and paired and DBS 91 samples were collected in the same manner from HBsAg carriers, and plasma was immediately stored at −80 °C. The 91 DBS samples were air-dried and stored in a zip-lock plastic bag with desiccant at −80 °C on the same day. Note that all samples were collected in the Medical Research Council Liver Clinic (MRC@LSHTM) in Fajara, The Gambia.

#### 2.1.2. DBS Optimization: Limit of Detection

Assessment of elution volume and detection limit of DBS for the study were carried out using plasma with known viral loads categorized as high (3.0–4.9 log_10_ IU/mL), medium (2.0–2.9 log_10_ IU/mL), low (1.0–1.9 log_10_ IU/mL) or undetectable (<1.0–0.0 log_10_ IU/mL) compared to DBS samples. For these, 3, 4 or 6 DBS punches prepared from 50, 75 or 100 µL of blood, respectively, were eluted in normal saline with a 1 in 10-fold dilution factor (for the latter two concentrations) or 1 in 13-fold factor (for former concentration). Henceforth, these samples will be referred to as sample 1 for 50 µL, sample 2 for 75 µL and sample 3 for 100 µL, respectively. All samples were vortexed, incubated at room temperature for 30 min and subsequently transferred into the cartridge to be analyzed on the GeneXpert platform. Following optimization, in all groups, three spots of 6.0 mm (1/8 inch) of DBS were placed in a 1.5 mL Eppendorf tube and eluted in 0.65 to 1.0 mL of normal saline. The mixture was vortexed and incubated at room temperature for 45 min to 2 h with gentle shaking.

### 2.2. Plasma

For freshly collected samples, 0.65 to 1.0 mL was added to pre-labeled HBV GeneXpert cartridges (containing all reagents required for HBV DNA quantification) and loaded on a 4-module GeneXpert system. For the stored samples, HBV DNA was quantified using three assays: an in-house real-time qPCR, GeneXpert HBV viral load (Cepheid AB, Röntgenvägen 5, SE- 171 54 Solna, Sweden) and COBAS TaqMan Roche (CAP-CTM; Roche Molecular Diagnostics, Basel, Switzerland). Samples from CHB carriers (n = 283) were used to isolate HBV DNA from 200 µL EDTA-plasma using the Qiagen Mini kit for in-house assay according to the manufacturer’s instructions (Qiagen GmbH, Qiagen Strasse 1, 40724 Hilden, Germany). The in-house qPCR analysis was performed as previously described [31]. GeneXpert HBV DNA viral load quantification (n = 306) was performed according to the manufacturer’s recommendation. In addition, serum samples (n = 244) were analyzed using the COBAS TaqMan Roche automated assay (Dakar, Senegal) according to the manufacturer’s instructions.

### 2.3. TAT and GeneXpert Mapping

The Gambia is divided into six administrative regions, and in each region, at least one GeneXpert machine is installed (Appendix A). The study thus encompassed the hospitals, health centers, clinics and laboratories from all the regions. These had varying numbers of machines, with the highest being in the greater Banjul region (n = 8). The purchasing of Cepheid cartridges in The Gambia for nationally owned health facilities was carried out by the funders of projects who either initially purchased the GeneXpert machine or are primary funders. The projects include funding agents like the Global Fund, TB, HIV, COVID-19 and the Ministry of Health of The Gambia. Each health facility obtained supplies from government-owned central stores. For the research center (Medical Research Council at LSHTM), purchases were made directly from Cepheid through its supply department. The study distributed a 23-question form to assess GeneXpert use and related challenges. It addressed a range of points, including location, machine use, purchasing, funding and maintenance of the GeneXpert machines across the country. The questionnaires were completed from the 18 April 2023 to the 2 May 2023, over the phone, during face-to-face meetings, and through emails where applicable. In addition, TAT was assessed in newly enrolled patients (n = 54) between March 2023 and April 2023 in the MRC liver clinic (MRC@LSHTM) from the time of HBsAg testing and sample collection dates for patients (n = 54). Also, qPCR viral load data from the Management and Treatment of Chronic Hepatitis B (MATCH-B) patients (n = 36/591 with viremia) analyzed between 2019 and 2020 were also included in the analysis of TAT as a control.

### 2.4. Data Analysis

The correlation between GeneXpert HBV assay and the level of HBV viremia detection was evaluated by Pearson’s correlation using the R studio. In addition, a Bland–Altman analysis (scatter plots of the differences between the paired measurements plotted against their means) of positive samples obtained by the two assays was used to visualize and assess graphically the magnitude of disagreement between them and estimate the overall bias of plasma and DBS samples using RStudio software (version 1.4869). Similarly, TAT and GeneXpert mapping were calculated using simple descriptive analysis.

## 3. Results

### 3.1. DBS Optimization: Limit of Detection (LOD)

DBS samples 1, 2 and 3 were prepared from 50, 75 and 100 µL blood, respectively, and eluted in 650, 750 and 1000 µL of saline solution. The GeneXpert HBV DNA test detected quantifiable viral loads in all three samples between 1.0 and 3.24 log_10_ IU/mL, corresponding to plasma samples with known loads of 1.0 to 4.74 log_10_ IU/mL (Table 1). Interestingly, samples 1 and 3 had a high number of undetected viral load, at one-half (50%) compared to plasma at one-quarter (25%). There was no difference in the number of samples with detectable viremia between samples 2 and 3.

### 3.2. Plasma and DBS HBV Viral Load—Group 1

Of the 56 paired plasma and DBS samples analyzed, 51 (91.1%) and 33 (58.93%) had quantifiable viral loads greater than 20 IU/mL, respectively, detected using the GeneXpert HBV viral load test. In addition, four plasma samples and 15 DBS samples had viral loads of under 10 IU/mL. The GeneXpert HBV assay showed a high correlation between plasma samples and DBS, with three plasma samples and eight DBS samples having an undetectable viremia, but there were no invalid (meaning the results were within the internal Quantitative Standard High and Low range, IQS-H or IQS-L) or erroneous results recorded, which indicates that no reagent problem was detected during the assay (Table 2). The linear regression of DBS against plasma showed a maximum measurement of 7.94, a minimum of 1 log_10_ IU/mL and a mean difference or bias of −1.413 (at a 95% detection limit of −1.588 to −1.241) with a correlation of r = 0.88, indicating a strong correlation between the assays (Figure 1).

### 3.3. Plasma HBV Viral Load—Group 2 

Study participants were mainly asymptomatic individuals classified as HBeAg-negative for chronic infection (n = 306). According to disease status, only 5% (n = 14) of patients had cirrhosis based on liver stiffness measurement or liver histology. The majority were male (71%), and among all participants, 40% were in their thirties and about 30% in their forties (Table 3). Among the 306 samples tested, 23 were excluded from the analysis because they could only be tested using the GeneXpert HBV. The differences in the limit of detection for qPCR (50 IU/mL), COBAS TaqMan (26 IU/mL) and GeneXpert (10 IU/mL), as well as the results obtained with each assay, are summarized in Table 4.

### 3.4. GeneXpert Against in-House qPCR

Of the 306 plasma samples analyzed, 249 (81.3%) had quantifiable HBV viral load, and 57 were undetectable. Among the samples with detectable viral loads, 109 samples had quantifiable viral loads detected by in-house PCR (Table 4). The assays show variation in detection levels of HBV DNA viremia, which could be associated with the following: (1) the lower limit of detection for each assay was ~5 IU/mL (in-house), 3.2 IU/mL (GeneXpert) and 20 IU/mL (COBAS TaqMan) and (2) the volume of sample required for the detection of HBV DNA by GeneXpert is 3 times that of the in-house assay. The discordance in log_10_ IU/mL with arithmetic values was observed in seven samples. Additionally, nine samples were undetected by the in-house test but had ≥3 log_10_ IU/mL viremia by the GeneXpert assay. The rest of the samples (n = 109/242) with arithmetic values were almost identical. Figure 2A shows the comparison between HBV DNA levels measured with the GeneXpert HBV viral load test and in-house qPCR. A strong concordance between the two assays was found with a correlation coefficient (r) of r = 0.90; Y = 1.0764x + 0.9540. The Bland–Altman plot analysis showed an average viral load between 1.352 and 8.033 Log_10_ IU/mL. The mean bias was determined at −1.173 log_10_ IU/mL, indicating that the in-house assay measures, on average, DNA levels 1.173 log_10_ IU/mL lower than the GeneXpert. This is almost statistically significant, with 95% limits of agreement observed (bias: −2.474 to +0.128 log_10_ IU/mL) (Figure 2B). The bias was shown to be smaller at lower viral loads compared to higher viremia. A few samples—4.59% (5/109)—fell outside the limits of agreement (Figure 2B).

### 3.5. GeneXpert Against COBAS TaqMan Assays

Among the 244 samples tested with the COBAS TaqMan assay (Roche), 129 samples had detectable HBV DNA, and 115 samples were undetectable. Among samples with detectable viral loads, 110 were quantifiable using the GeneXpert HBV test. These samples were used in the statistical analysis, which led to a correlation coefficient of r = 0.89 and Y = 1.227x − 1.2629 (Figure 3A). As shown by the Bland–Altman plot of GeneXpert- minus COBAS TaqMan qPCR-measured plasma viral loads, the mean difference (bias) observed was 0.316 log_10_ IU/mL. The bias resulted in a weak correlation between the two assays at 95% limits of agreement (bias: −1.188 to +1.820 log_10_ IU/mL), with 5.45% (6/110) found outside of the limits of agreement. Overall, the average viral load ranged from 1.445 to 8.485 log_10_ IU/mL (Figure 3B).

### 3.6. In-House qPCR Versus COBAS TaqMan for Plasma Viral Load

Of the 116 quantifiable samples by in-house qPCR and 129 samples by COBAS TaqMan assay with numerical results, 79 samples had detectable viral load according to the two assays. A scatter plot of in-house qPCR versus COBAS TaqMan for these viral loads showed a correlation of r = 0.89 with a Y = 1.0843x − 1.800 (Figure 4A). The plot showed a higher correlation for samples with high viremia than those with lower viral load. The Bland–Altman plot showed that 6.33% were outside the limits of agreement, with a mean bias or difference of 1.421. The bias was statistically significant, with 95% limits of agreement of +0.129 to +2.713 log_10_ IU/mL. The viral load mean was between 2.358 and 7.853 log_10_ IU/mL (Figure 4B).

### 3.7. Analysis Using Plasma GeneXpert Against DBS GeneXpert

For the stored samples, of the paired plasma and DBS (n = 91) samples analyzed with the GeneXpert HBV test, 37 had detectable HBV DNA, 18 undetectable viral load and 33 (33.6%) invalid results (Table 5). Overall, kappa statistics (ĸ) were 0.10, indicating a very poor concordance between the results. Even after excluding samples with errors or invalid results for at least one type of sample (n = 51), kappa statistics (ĸ) were low (0.34), indicating a poor correlation.

The scatter plot analysis of 34 GeneXpert plasma and DBS samples with viral loads showed a correlation of r *=* 0.88 and Y = 0.7014x − 0.7420 log_10_ IU/mL (Figure 5A). The Bland–Altman statistical analysis showed that 11.76% of samples fell outside the limits of agreement, with a mean difference of 1.831 Log_10_ IU/mL. The difference was significant at 95% limits of agreement +0.660 to 3.001 Log_10_ IU/mL, and the bias was smaller for samples with lower viral loads (Figure 5B) (Table 5).

### 3.8. Turn-Around Time, GeneXpert Mapping and Real-World Challenges

Patients (54/138) newly enrolled and whose samples were analyzed using the GeneXpert HBV test were assessed for (TAT) from HBsAg testing, DNA amplification and initiation of treatment. Of the 36/591 with viral load >200,000 IU/mL of the control enrolled in treatment, 24/36 were included to determine TAT from testing to treatment initiation. After the analysis, 3/54 freshly collected plasma and DBS samples were eligible for treatment within six days of enrollment for this study (Table 6). Those eligible for treatment during this period would be on surveillance for between 3 to 6 months.

A total of 20 questionnaires were completed from 17 facilities (attached; appendices). The TAT for ordering and receiving HIV and TB GeneXpert cartridges from the central stores varied across the national health facilities. National health facilities reported a good TAT, with a mean time of 12 days. In addition, TATs were reported to be impacted by limited transportation and supply chain issues, including shipping logistics; rare country-wide shortages; and limited funding to purchase assay-specific cartridges.

The results obtained from the survey revealed that the challenge for all facilities includes loss of cartridges due to error runs and invalid results generated by the GeneXpert machines. These rates per week account for 20% (one in five cartridges) (Table 7). Other challenges include low electricity voltage and improper infrastructure (e.g., non-air conditioning, storage space), which were indicated as barriers to the installation and low usage. In terms of maintenance, both in-house and external biomedical teams were reported across all facilities with an average TAT of 9 days for the repair of a machine during breakdown. Of note, three of the health facilities reported no major breakdowns at the time of the survey.

Close to 80% of machines were reported to function in dust-free rooms, 90% in air-conditioned rooms and 10% reported non-air-conditioned rooms. In addition, 90% of the machines were connected to a UPS system. This served as a backup for the electricity supply. Moreover, all machines were well maintained and underwent regular maintenance. Finally, the research facility, where purchasing is done internationally, reported a TAT between 4 and 12 weeks.

A total of 20 GeneXpert machines were reported across the country; these were installed between 2015 and 2022, except for two that are currently not installed. Additionally, 65% of the machines were relatively new and installed between 2021 and 2022, and 85% of the machines were funded through programs or projects, with 60% of the machines across the country funded by the Global Fund. Distribution of the GeneXpert machines varied across the country, with at least one GeneXpert machine in each region. However, the majority (55%) of the machines were in the Greater Banjul area. A total of 84 staff across all the health facilities are currently trained to operate the GeneXpert systems. GeneXpert HBV viral load analysis was reported in only one facility (MRCG). The machines in national facilities are mainly used to test for HIV, COVID-19, TB and Early Infant-HIV, while in the research center, in addition to the abovementioned assays, HBV diagnosis is also carried out.

## 4. Discussion

This is the second study conducted in sub-Saharan Africa to determine the diagnostic accuracy of the GeneXpert HBV viral load test. We showed that GeneXpert HBV Viral Load could accurately quantify HBV DNA in plasma and has the potential to do the same in DBS. We found excellent concordance and diagnostic accuracy for the detectable HBV viral loads assessed by the GeneXpert HBV viral load test and the gold standard assays for freshly collected and stored plasma and DBS samples, with strong correlation coefficients of r = 0.9. The GeneXpert HBV test had the highest number of quantifiable viral loads (81.3%) for plasma samples. The Bland–Altman plot limit of agreement between GeneXpert HBV and COBAS TaqMan had a mean bias of 0.316 log_10_ IU/mL and depicted a stronger correlation compared to GeneXpert vs. the in-house assay and COBAS TaqMan vs. the in-house assay, which had a mean bias of −1.173 and −1.421 log_10_ IU/mL, respectively, for plasma samples. However, the Bland–Altman plot for paired stored plasma and DBS samples had a low number of samples with detectable viremia (n = 34), with a smaller (for lower viral loads) mean bias of 1.831 log_10_ IU/mL, which is almost significant at 95% limits of agreement: 0.66–3.001. The study successfully mapped GeneXpert machines in the six regions, providing an excellent opportunity for scaling up HBV screening and monitoring in The Gambia.

A number of studies have compared the diagnostics accuracy of GeneXpert HBV DNA Viral Load assay in fresh plasma and DBS samples in more developed countries but only a few have done so in the sSA. According to the manufacturer, this assay is highly sensitive, as it can detect viral loads as low as 10 IU/mL and is highly specific if performed a few hours after blood sampling. Similarly, the GeneXpert HBV cartridge has an excellent detection limit for plasma EDTA and serum (5.99 IU/mL), with a reproducibility range of ≤0.28 log_10_ [17,18]. In addition, it is easy to use and presents a remarkable TAT, with results available in between 1 and 2 h [32,33].

The results were highly accurate in 91.1% of freshly collected plasma, DBS and stored plasma samples. The concordance between the HBV DNA viral load generated in qPCR and GeneXpert HBV assays, GeneXpert HBV viral load and qPCR during optimization was excellent. This corroborates results obtained by Jackson et al. [34], who analyzed different blood volumes prepared with 50 and 75 μL of DBS samples (stored for six months) and found a decline in the viral load between 0.18 and 0.66 log_10_ IU/mL.

The new HBV Viral Load assay compared to the in-house qPCR based on Sybr Green dye has so far not been performed, to the best of our knowledge. The in-house qPCR assay has been the gold standard for HBV DNA viral load detection in The Gambia for more than a decade, as it is known to be cheap and highly sensitive, with the capacity of detecting viral loads as low as 30 copies/mL^−1^ [31]. The diagnostic accuracy of GeneXpert HBV versus in-house qPCR (Sybr Green) viral load measured with sera showed a high concordance of 90%, which is almost significant in the Bland–Altman plot range. On the other hand, studies comparing GeneXpert to commercial assays found higher concordance to Aptima Quant [19], CAP/CTM HBV v2 [20] and COBAS Ampliprep/TaqMan Roche [13]. These commercial HBV DNA assays have been highly standardized compared to the in-house qPCR. They are closer to the GeneXpert HBV test in that they are both well calibrated and have undergone quality controls before being commercialized, which is not the case for in-house qPCR. Above all, GeneXpert has several advantages, including the enclosed cartridge system with controls and standards, which has minimal contamination issues, lower assay-to-assay variation, and it is cheaper, easy to implement and requires much less sample preparation time (about 10 to 20 min).

There are a few studies that evaluated the diagnostic accuracy of the GeneXpert assay for HBV DNA quantification compared to COBAS TaqMan assays in real-life settings [14,17,35]. Marcuccilli et al. [17] compared the performance of GeneXpert HBV to CAP/CTM, COBAS^®^ AmpliPrep/COBAS^®^ TaqMan HBV test and used the COBAS TaqMan real-time HBV test, though there is a discrepancy in confirming the correlation on samples from three European countries (Italy, France and Germany) and the United States. Similarly, Gupta et al. [21] compared the GeneXpert HBV and the COBAS TaqMan real-time test using samples from chronic carriers who were less than 18 years of age in India. However, Woldemedihn et al. conducted the first study that used stored samples to validate the GeneXpert HBV test in a real-life setting in sSA [14]. Our study is the second to compare the GeneXpert HBV Viral Load kit vs. COBAS TaqMan in stored samples in the sSA setting. It thus provides important insights into the performance of the GeneXpert compared to the COBAS Taman test using both stored plasma and DBS samples. Our results show an excellent correlation, but Bland–Altman plots indicate a small bias between these assays, which is almost significant at the 95% limits of agreement. The strong concordance in this study is comparable to that of a study using samples from similar origin, with a correlation of 0.23 log_10_ IU/mL [14]. In another setting, a study conducted in India looked at the concordance between assays analyzed using Bland–Altman plots and revealed that they were within the range of −1.13 to 1.1 log_10_ IU/mL, with a lower bias of −0.018 log_10_ IU/mL [35].

DBS is a sample collection technique best suited for resource-limited countries and marginalized populations with low access to care, which could also support the WHO’s objectives for HBV elimination by 2030 [36]. The combined use of plasma and DBS samples shows high diagnostic accuracy and strong correlation for both freshly collected and stored plasma and DBS paired samples. The data obtained in this study demonstrated that the GeneXpert HBV Viral Load could accurately quantify HBV DNA in plasma and has the potential to do the same in DBS. Although the use of DBS is not directly recommended by Cepheid, Poiteau et al. [19] and Bargin et al. [36], our study evaluated the diagnostic accuracy of the Xpert HBV test by measuring whole-blood specimens from stored DBS and found that this sample type had 2 log_10_ lower viral load compared to the plasma specimens. The 91 paired plasma–DBS stored samples that were analyzed (group 2) showed a good correlation coefficient with a small bias. This bias was statistically significant for samples with lower viral loads, indicating that the GeneXpert assay has the potential to detect HBV DNA in whole blood DBS specimens, especially those with high viral loads above log_10_ 2. Although the GeneXpert generated invalid results, the difference could be due to the low viral load, given its threshold of HBV DNA detection.

## 5. Conclusions

Our data showed that the GeneXpert HBV viral load test is highly sensitive compared to the gold standard assays and accurate in quantifying HBV DNA in both plasma and DBS fresh or stored samples from sub-Saharan African origins. This test has a short TAT, as assays can be completed within an hour and do not require pre-assay preparation and post-assay data analysis. This accurate, standardized and simplified use should increase the scaling-up of HBV diagnosis, monitoring and treatment in resource-limited settings. Other HBV DNA quantification techniques are expensive and only available in urban cities.

Even though DBS samples led to more invalid/erroneous results when dealing with very low viral loads compared to other sampling techniques, DBS could be used for patient monitoring and treatment enrollment since it may identify patients with viral loads below the cut-off viremia of 2000 IU/mL as per the EASL treatment initiation guideline.

Access to the GeneXpert platform across the country could offer new perspectives on scaling up screening in order to identify asymptomatic chronic hepatitis B carriers in The Gambia.

To this end, the research projects in The Gambia could use these platforms for patients residing in rural areas to manage and monitor their disease burden. One of the major barriers to DNA testing in the sSA is equipment and trained staff to operate them, but with the availability of 20 GeneXpert machines and 84 trained staff, that could easily be mitigated. A better collaboration with the hospitals, health centers and clinics could further improve the hepatitis B projects in The Gambia. In addition, reaching out to Cepheid, Global Fund and the Ministry of Health of The Gambia could be beneficial since they may only need to fund software installation, cartridges and the expansion of existing projects. This may provide the opportunity to screen and treat people, resolving issues linked to nucleic acid testing in sSA.

## Figures and Tables

**Figure 1 microorganisms-12-02273-f001:**
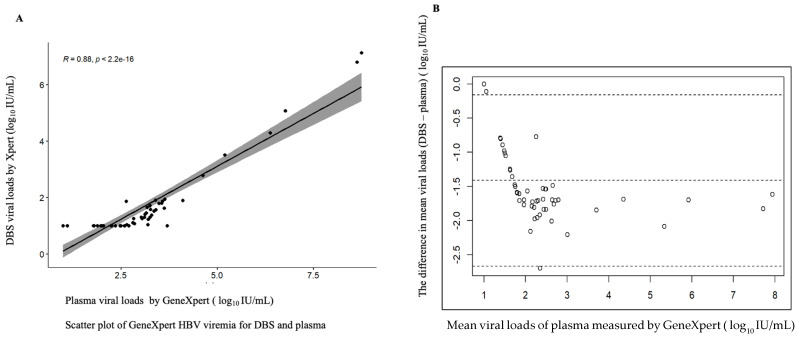
Correlation between DBS and plasma of fresh samples. (**A**) Scatter plot of DBS and plasma viral loads. (**B**) Bland–Altman plot of the mean difference of plasma minus DBS viral loads. The correlation coefficient r = 0.88, the bias is −1.4 and 3/56 falls outside of the limits of agreement; the maximum measurement lies between 1 and 7.94 log 10 IU/mL.

**Figure 2 microorganisms-12-02273-f002:**
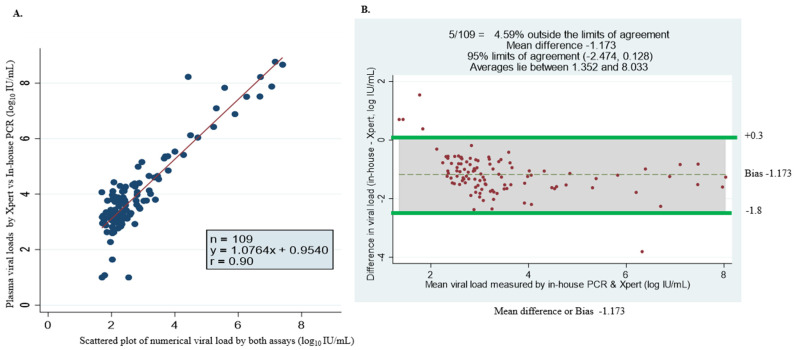
Assay comparison analysis using linear regression methods and Bland–Altman plot of HBV DNA viral load levels measured by GeneXpert HBV viral load and in-house PCR assay. (**A**) Simple linear regression (scatter plot) of 109 specimens with numerical viral load quantified by both assays. (**B**) Bland–Altman plot of 109 serum samples with detectable viral load by GeneXpert- minus in-house PCR-measured plasma viral load (vertical axis) against mean of GeneXpert- and in-house PCR-measured plasma viral loads (horizontal axis); the data (dotted) represent mean differences of −1.173 at limits of agreement at 95% CI of −2.474 and +0.128, average viral load is between 1.352 and 8.033 log_10_ IU/mL and 4.59% (5/109) of the data are found outside the limits of agreement.

**Figure 3 microorganisms-12-02273-f003:**
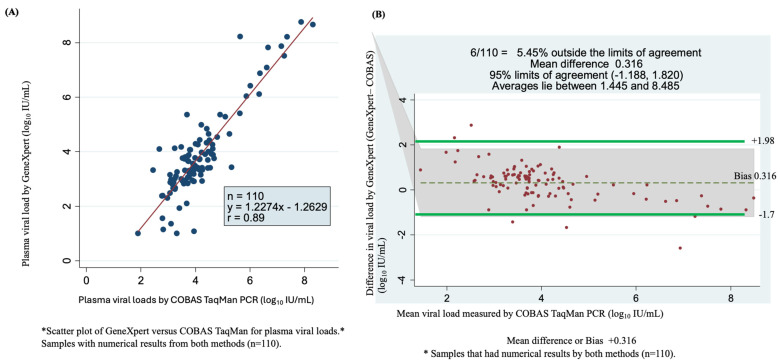
Assay accuracy and concordance determined using scatter plots and Bland–Altman plots of detectable viral load by GeneXpert HBV DNA viral test and COBAS TaqMan PCR. (**A**) Comparison of the sensitivity and correlation of 110 serum samples of numerical results for the two assays using a scatter plot. (**B**) Bland–Altman plot of 110 serum samples viral load by GeneXpert minus COBAS TaqMan PCR measured plasma of quantifiable viral load (vertical axis) against mean of GeneXpert and COBAS TaqMan PCR plasma measured viral loads (horizontal axis), the dotes represent mean difference of +0.316 at 95% limits of agreement of 1.188 to 1.820, an average is between 1.44 and 8.485 log_10_ IU/mL and 5.45% (6/110) of samples are found outside the limits of agreement.

**Figure 4 microorganisms-12-02273-f004:**
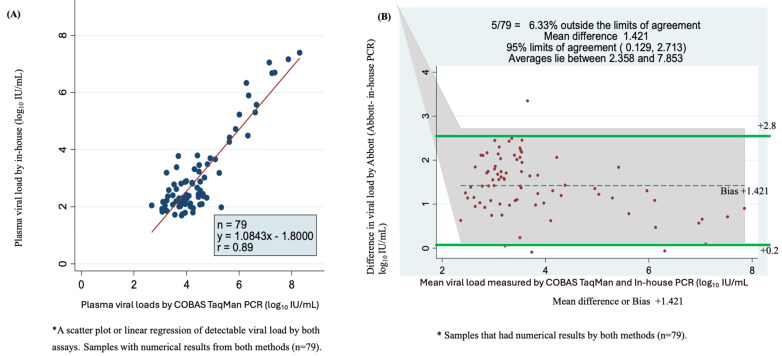
Accuracy, correlation and sensitivity of the two assays using plasma sample viral loads using scatter plot or linear regression and Bland–Altman plot. (**A**) Scatter plot of 79 serum samples quantified by COBAS TaqMan and in-house PCR test. (**B**) Bland–Altman plot analysis of 79 samples with detectable viral loads of in-house PCR minus COBAS TaqMan PCR (vertical axis) against the mean of the in-house PCR- and COBAS TaqMan PCR-measured plasma viral loads (horizontal axis); the data represent the mean difference of +1.421 at 95% limits of agreement of 0.129 to 2.713, the average lies between log_10_ IU/mL 2.358 and 7.853 and 6.33% (5/79) are found outside the limits of agreement.

**Figure 5 microorganisms-12-02273-f005:**
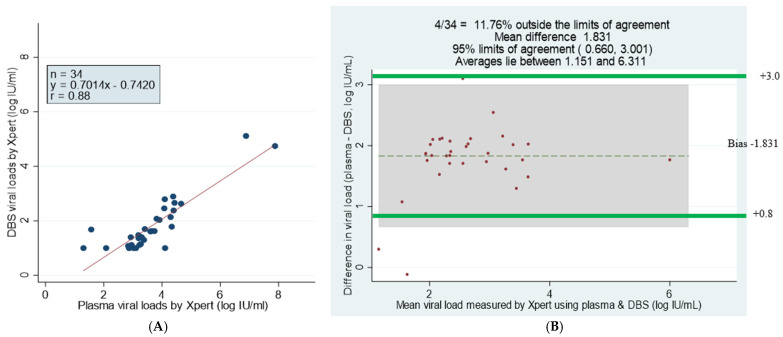
Comparison and correlation of DBS and plasma paired samples using scatter and Bland–Altman plots of detectable viral loads by GeneXpert HBV DNA test. (**A**) Linear regression of 34 paired DBS–plasma samples with detectable viral loads by GeneXpert. (**B**) Bland–Altman plot analysis of GeneXpert viral load of DBS minus plasma viral loads (vertical axis) against GeneXpert viral load mean of DBS and plasma viral loads (horizontal axis); mean difference of 1.831 at limits of agreement of 0.660 to 3.001; the average lies between log_10_ IU/mL 1.151 and 6.311 and 11.76% of the data are outside of the limits of agreement. Samples that had numerical results by both methods are presented (n = 34).

**Table 1 microorganisms-12-02273-t001:** Summary of batch 5 (group 1) samples of fresh plasma and DBS samples used for optimization.

Batch 5		
Sample 1 = 50 μL—3 Punches (650 μL Elution)	DBS Viral Load	Plasma Viral Load
EG5894	1.73 × 10^3^ IU/mL log 3.24	5.47 × 10^4^ IU/mL log 4.74
EG5892	<10 IU/mL log 1.00	276 IU/mL log 2.44
EG5102 V2	HBV NOT DETECTED	<10 IU/mL log 1.00
EG5095 V2	HBV NOT DETECTED	HBV NOT DETECTED
	Sample 2 = 75 μL—4 Punches (750 μL Elution)	
EG5894	1.75 × 10^3^ IU/mL log 3.24	5.47 × 10^4^ IU/mL log 4.74
EG5892	<10 IU/mL log 1.00	276 IU/mL log 2.44
EG5102	<10 IU/mL log 1.00	<10 IU/mL log 1.00
EG5095	HBV NOT DETECTED	HBV NOT DETECTED
	Sample 3 = 100 μL—6 Punches (1000 μL Elution)	
EG5894	2.07 × 10^3^ IU/mL log 3.32	5.47 × 10^4^ IU/mL log 4.74
EG5892	12 IU/mL log 1.08	276 IU/mL log 2.44
EG5102	HBV NOT DETECTED	<10 IU/mL log 1.00
EG5095	HBV NOT DETECTED	HBV NOT DETECTED

Abbriviations: HBV, hepatitis B virus; IU, international unit, EG, enroll Gambia; DBS, dried blood spot.

**Table 2 microorganisms-12-02273-t002:** Comparison of HBV concentrations in GeneXpert HBV Viral Load plasma and DBS for group 1.

	GeneXpert HBV Viral Load Plasma (No. of Plasma)	GeneXpert HBV DBS Viral Load (No. of DBS)
Not detected	3	8
<10 IU/mL	4	15
≥20 IU/mL	51	33
Total	58	56

Two of the DBS samples were excluded from the analysis due to error runs with no samples remaining for a third repeat. Abbreviations: HBV, hepatitis B virus; DBS, dried bloof spot.

**Table 3 microorganisms-12-02273-t003:** Characteristics of study participants for the stored samples (n = 306, group 2).

Variables		Number (%)
Sex	Men	202 (71%)
	Women	81 (29%)
Age (years)	18–29	23 (8%)
	30–39	120 (43%)
	40–49	83 (29%)
	≥50	57 (20%)
Disease status (by USA)	No apparent liver disease	261 (92%)
	Cirrhosis	14 (5%)
	HCC	1 (1%)
	Fatty liver disease	7 (2%)

Abbreviations: USA, united state of america; HCC, hepatocellular carcinoma.

**Table 4 microorganisms-12-02273-t004:** HBV DNA testing using plasma samples for the different assays.

GeneXpert	In-House PCR		COBAS TaqMan	
	Undetectable HBV DNA (n = 167)	Detectable HBV DNA (n = 116)	Undetectable HBV DNA n = 137	Detectable HBV DNA n = 117
Undetectable	39 (23%)	0 (0%)	0 (0%)	0 (0%)
Detectable	120 (72%)	109 (94%)	129 (85.3%)	110
Error	4 (2%)	3 (3%)	4 (2%)	3 (3%)
Invalid	4 (2%)	4 (3%)	4 (2%)	4 (3%)

Abbreviations: PCR, polymerase chain reaction; HBV, hepatitis B virus; DNA, deoxyribonucleic acid; Error = aborted assays due to sample volume, probe integrity problem; invalid = sample outside of detection limit of the internal quantification standards.

**Table 5 microorganisms-12-02273-t005:** HBV DNA viral loads for the paired plasma and DBS samples using GeneXpert.

		Plasma Samples			
		Undetectable (n = 14)	Detectable (n = 73)	Error (n = 2)	Invalid (n = 2)
**DBS samples**	Undetectable (n = 18)	5 (36%)	11 (15%)	1 (50%)	1 (50%)
	Detectable (n = 37)	1 (7%)	34 (47%)	1 (50%)	1 (50%)
	Error (n = 3)	0 (0%)	3 (4%)	0 (0%)	0 (0%)
	Invalid (n = 33)	8 (57%)	25 (34%)	0 (0%)	0 (0%)

n, number of patients; DBS, dried blood spot; Error, aborted assays due to sample volume, probe integrity problem; invalid= sample outside of detection limit of the internal quantification standards.

**Table 6 microorganisms-12-02273-t006:** Turnaround time between GeneXpert and qPCR for the research clinic.

Median TAT	GeneXpert	qPCR
From HBsAg testing to enrollment to the study	8 days (IQR (8–12.5))	N/A
Sample collection to HBV DNA testing completion	138 min (IQR 113–179)	83 days (28–122)
HBV DNA results to treatment initiation	6 days (IQR 5.5–6.5)	18 days (16–37)

qPCR treatment initiation data for TAT analysis was excluded for 12 patients who started treatment on enrolment day or before qPCR viral load analysis. qPCR, quantitative polymerase chain reaction; TAT, turn-around-time; HBV, hepatitis B virus; DNA, deoxyribonucleic acid; IQR, interquartile range.

**Table 7 microorganisms-12-02273-t007:** Summary of the key responses from the questionnaire about GeneXpert use in health facilities.

Regions	No. Health Facilities	GeneXpert Module (s) and Cartridge No	Purpose (Assays)	Funders	Cartridge Loss (Error or Invalid)/Average Per Week
West Coast	3	4 (1), 16 (2)	HIV, TB, and early infant diagnosis	Global Fund, USA Army	6
Greater Banjul	8	4 (8), 16 (2) and infinity (1)	COVID-19, HIV, TB, HBV, early infant diagnosis, and host response 3-gene research	Global Fund	6
Central River	2	4 (2)	COVID-19, HIV, TB, and early infant diagnosis	Global Fund	10
Upper River	1	4 (1)	COVID-19, HIV, TB	Global Fund	5
North Bank	2	4 (2)	COVID-19, HIV, TB and early infant diagnosis	Global Fund, UNICEF	5
Lower River	1	4 (1)	HIV, TB and early infant diagnosis	Global Fund	5

HIV, human immunodeficiency virus; TB, Tuberculosis; USA, United State of America, HBV, hepatitis B virus; COVID-19, coronavirus disease; UNICEF, United Nations International Children's Emergency Fund.

## Data Availability

The original contributions presented in the study are included in the article and Appendix A, further inquiries can be directed to the corresponding authors.

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
