# Peer review of "Real-Life Diagnostic Accuracy and Clinical Utility of Hepatitis B Virus (HBV) Nucleic Acid Testing Using the GeneXpert Point-of-Care Test System from Fresh Plasma and Dry Blood Spot Samples in The Gambia"

_microorganisms, 2024, doi:10.3390/microorganisms12112273_

Round 1
Reviewer 1 Report
Comments and Suggestions for Authors
This study analyzed the accuracy of GeneXpert HBV test. This simplified system may be useful especially in African HBV endemic countries. However, there are several concerns. This manuscript is very difficult to read due to lack of sufficient information and typographical errors. And discussion with interpretation of the results is insufficient.
Major comments
1. Of the 306 plasma samples, 249 were HBV positive by Xpert test. And among these 249 samples, only 109 samples were HBV positive in-house PCR (gold standard) (Section 3.4). The results of Xpert and in-house PCR were quite different. These results may mean that Xpert test is more sensitive, but it may also mean that false positives were detected by Xpert.
Additionally, the HBV titers by Xpert and the titers by in-house PCR were correlated well, but there is also a bias of -1.173 log10 IU/mL. This indicate that resulting titers has 10-fold difference. Why do the authors conclude that Xpert is accurate, even though it differs significantly from the gold-standard results? What does Xpert positive but PCR negative samples mean? Please state clearly with supporting evidence.
2. Please describe the cost of Xpert in Introduction section.
3. In Result 3.1, please describe the optimized method (3 spots with 0.65-1 mL elution buffer?), and why the authors decided to do so.
4. Please describe the samples and study participants in detail. How many people participated? Are all of them infected with HBV? By what method were they diagnosed?
5. In all Figures, describe what the x-axis shows and what the y-axis shows. It is very difficult to understand.
Minor comments
6. Please describe how the authors calculate the limits of agreement in Bland-Atman plot.
7. In page 4 line 33, what does “NS” mean? Normal saline? I consider some abbreviations, including NS, are not necessary. It is difficult to read due to many abbreviations.
8. In page 6 line 36, “Table 2” here is not correct.
9. In page 7 line 82, please correct “(TAT)”.
10. In Table 1, please correct “100 uL elution” to “1000 uL elution”.
11. In Result 3.2, 56 samples were analyzed, but in Table 2, the total number of serum samples were 58. What is the difference between these numbers?
12. Why is the sample number 306 but the total number of study participants in Table 2 is 283? Were 306 serum samples from 283 participants used in this study? If so, please describe clearly.
13. In Table 4 and 5, what do “Error” and “Invalid” mean? Please state in detail.
14. Table 7 is difficult to understand. Please separate columns GeneXpert and qPCR. IQR should also be consistent (25 percentile – 75 percentile). Please illustrate the work of GeneXpert and qPCR.
Author Response
Reviewer 1
This study analyzed the accuracy of GeneXpert HBV test. This simplified system may be useful especially in African HBV endemic countries. However, there are several concerns. This manuscript is very difficult to read due to lack of sufficient information and typographical errors. And discussion with interpretation of the results is insufficient.
Major comments
- Of the 306 plasma samples, 249 were HBV positive by Xpert test. And among these 249 samples, only 109 samples were HBV positive in-house PCR (gold standard) (Section 3.4). The results of Xpert and in-house PCR were quite different. These results may mean that Xpert test is more sensitive, but it may also mean that false positives were detected by Xpert. Thank you for your observation. We added some more that was not included in page 7 line 40-43. Sorry about this.
Additionally, the HBV titers by Xpert and the titers by in-house PCR were correlated well, but there is also a bias of -1.173 log10 IU/mL. This indicate that resulting titers has 10-fold difference. Why do the authors conclude that Xpert is accurate, even though it differs significantly from the gold-standard results? What does Xpert positive, but PCR negative samples mean? It is well known than even between 2 commercialized assays they are discrepancies of such order, here we compare an “In house “ assay with its own limitations even if it was very usefull for the last 10 years with a new standart, automatized assay and that allow to link to care positive patients that enter the international criteria. Thank you for your comment, this is very useful. We have clarified this point with the English Language corrections that was carried out. Sorry about this.
- Please describe the cost of Xpert in Introduction section. Your comment is highly appreciated please see Page 2-line 39 to 45 to 46.
- In Result 3.1, please describe the optimized method (3 spots with 0.65-1 mL elution buffer?), and why the authors decided to do so. Thank you for your comment. The method was adapted from Jackson et al. 2021, please refer to reference no. 34.
- Please describe the samples and study participants in detail. How many people participated? Are all of them infected with HBV? By what method were they diagnosed?
- In all Figures, describe what the x-axis shows and what the y-axis shows. It is very difficult to understand. Thank you for your comments, this is much appreciated. All figures were amended accordingly.
Minor comments
- Please describe how the authors calculate the limits of agreement in Bland-Atman plot. Thank you for this insightful comment. Please refer to page 3 line 107 to 110.
- In page 4 line 33, what does “NS” mean? Normal saline? I consider some abbreviations, including NS, are not necessary. It is difficult to read due to many abbreviations. Thank you for your comment, all the abbreviations in relation to N.S were amended.
- In page 6 line 36, “Table 2” here is not correct. Thank you, this has been changed. Sorry about that.
- In page 7 line 82, please correct “(TAT)”. Thank you for this thoughtful comment. Please refer to line page 2 line23
- In Table 1, please correct “100 uL elution” to “1000 uL elution”. Thank you for the comment. It has been corrected accordingly. Sorry about that.
- In Result 3.2, 56 samples were analyzed, but in Table 2, the total number of serum samples were 58. What is the difference between these numbers? Thank you for the comment and I appreciate it. During the analysis 2 samples were excluded for the DBS samples due to insufficient samples volume. It is why we have this small discrepancy by numbers. Sorry about that.
- Why is the sample number 306 but the total number of study participants in Table 2 is 283? Were 306 serum samples from 283 participants used in this study? If so, please describe clearly. Thank you for spotting that out. Please refer to page 7 line 28, were it states that 23 samples were excluded during analysis.
- In Table 4 and 5, what do “Error” and “Invalid” mean? Please state in detail. Thank you for the brilliant suggestion. Please see below and page 7 line 16 and line 17-18 respectively.
Invalid: Presence or absence of the HBV DNA that cannot be determined, the IQS-H and/or IQS-L Cts are not within the valid range.
An ERROR: Presence or absence of the HBV DNA cannot be determined due to probe error or samples volume.
- Table 7 is difficult to understand. Please separate columns GeneXpert and qPCR. IQR should also be consistent (25 percentile – 75 percentile). Please illustrate the work of GeneXpert and qPCR. Thank you for the brilliant suggestion. The table has been changed accordingly.
Reviewer 2 Report
Comments and Suggestions for Authors
General comment
The impression from this paper is very ambivalent
First, the report on progress in the diagnosis of chronic hepatitis B virus infection in a highly endemic LMIC like The Gambia is relevant and in large parts novel. Furthermore, it provides interesting aspects on how a new sophisticated technique can be installed in such a country.
Unfortunately, the paper has many weaknesses.
a) Many papers have already reported on the performance of the GeneXpert HBV Viral Load test in a LMIC.
b) The text and the data contain many inaccuracies, inconsistencies and frank errors as pointed out below.
c) Furthermore, the language is often insufficient and partially not understandable.
These weaknesses can possibly be corrected in a revision.
Specific points
1. Title. Spell out HBV in title.
2. Abstract. L18. Unexplained abbrev. : TAT. Explain here also and not only later, or spell out here.
3. Abstract. L23. This sentence is not clear: “Xpert HBV test had the highest quantifiable HBV DNA viremia (n=249) and the lowest was detected by in-house qPCR (n=116) for stored plasma samples.” Percentages of positive results would be more informative.
4. Abstract. L25. Again, this sentence is not clear. Is 10 IU/mL the limit of detection or of quantitation?
5. Intro. L43. FDA in USA is well known, but what does TGA in Australia mean?
6. M & M. para. 2.1. How were the HBsAg carriers identified before they entered the PROLIFICA study?
a. Did they have symptoms of liver disease, were they blood donors or contact persons to HBV carriers?
b. How sensitive were the tests for identification and during study, i.e., the limit of detection for HBsAg?
c. Was HBsAg quantified?
d. Were HBeAg or HBcrAg results available?
7. M & M. L41, 47 and later. The Cobas Taqman test for HBV DNA is from Roche, not from Abbott.
8. Results. L5-10. This para. 2.1 is not well explained. It remains open which aliquot of plasma in the DBS samples with 3, 4, and 6 punches ended up in the Xpert test and how much plasma corresponded to amount of HBV DNA given into the Xpert test. Are the differences between DBS and plasma in the quantification only due to the different dilutions or different sizes of aliquots or is HBV DNA lost during elution of the DBS?
9. Results L11-20. The text should refer to the two following points or the paper should be corrected correspondingly.
a. Which kind of samples were tested? Were these all HBV DNA positive in previous highly sensitive tests?
b. Table 2. Why are 58 plasma samples in the table but only 56 for the DBS samples? If 2 results are missing in the DBS group, these two corresponding plasma samples should have been deleted.
c. Fig. 1B. Why does in this figure of the Bland-Altman plot the X-axis show the mean difference of plasma minus DBS viral loads whereas in all four other plots the mean value of the two variables is shown?
10. Results L22-78.
a. Table 4 and L26, 46, 47. Was the Cobas Taqman or the Abbott test used for HBV DNA? Cobas Abbott does not exist. The figure legends show “Abbott” and the discussion mentions in L61 also Abbott, but is this true?
b. L36. Table 3 should be corrected to table 4. The large differences in the proportion of positive results between the three tests would deserve some explanatory sentences in the main text.
c. L40-44. A brief explanation on what the “bias” practically means would be helpful. Does it mean that the in-house test measures on the average DNA levels by 1.173 log IU/mL lower than the Xpert test?
d. Fig.3 A & B. The letters at the Y-axis not readable and one of the two texts is probably wrong in both panels.
e. L 49. It is interesting that the Xpert test yields obviously slightly higher values (mean bias = 0.316) than the Abbott test (if I guess right from the confusing graphs) but 5 samples positive in the Abbott test were not detected with Xpert test.
f. L57-65. It must be cleared whether the commercial test was from Roche or Abbott.
g. Fig. 4B and L61-65. The figure shows clearly that the commercial test yielded in the average 1.421 log IU/mL higher values than the in-house PCR. However, the bias is positive with 1.421. Does this also mean that the in-house test was considered to be the reference test which would be not appropriate?
11. Results para. 3.8
a. L84. What happened with the 51 other patients with >200,000 IU/mL who were not eligible within 6 days? Did they receive a therapy later and how much later?
b. L97. How does the 55% rate per week fit to the 1 in 5 lost cartridges?
12. Discussion
a. L11. Shouldn’t the bias be -1.421? See point 10g.
b. L14. Is a bias of 1.831 small? Plasma gives 68 times higher IU/mL values than the DBS. This could decide whether therapy is planned or not.
c. L29. Do you mean with “assays” the commercial assays?
d. L30. A bias of -1.831 is not excellent concordance, because the HBV DNA level decides about therapy.
e. L82, 94, and 102. See above under 12 b). The DBS sample are inferior not only in sensitivity, but also for exact quantification, and that should be acknowledged. At least, the limitations should be discussed in more detail.
Comments on the Quality of English Language
The language is often insufficient and partially not understandable.
Author Response
Reviewer 2
Submission Date
09 September 2024
Date of this review
18 Sep 2024 09:39:48
Comments and Suggestions for Authors
General comment
The impression from this paper is very ambivalent
First, the report on progress in the diagnosis of chronic hepatitis B virus infection in a highly endemic LMIC like The Gambia is relevant and in large parts novel. Furthermore, it provides interesting aspects on how a new sophisticated technique can be installed in such a country.
Unfortunately, the paper has many weaknesses.
- a) Many papers have already reported on the performance of the GeneXpert HBV Viral Load test in a LMIC.
- b) The text and the data contain many inaccuracies, inconsistencies and frank errors as pointed out below.
- c) Furthermore, the language is often insufficient and partially not understandable.
These weaknesses can possibly be corrected in a revision.
Specific points
- Title. Spell out HBV in title. Thank you for the nice comment. Please refer to page 2 line 1-2
- Abstract. L18. Unexplained abbrev. : TAT. Explain here also and not only later or spell out here. Thank you for spotting that out. Please refer to page 1 line 11
- Abstract. L23. This sentence is not clear: “Xpert HBV test had the highest quantifiable HBV DNA viremia (n=249) and the lowest was detected by in-house qPCR (n=116) for stored plasma samples.” Percentages of positive results would be more informative. Thank you for the insightful suggestion. Please refer to page 1 line 24-26. Sorry about the oversight.
- Abstract. L25. Again, this sentence is not clear. Is 10 IU/mL the limit of detection or of quantitation? Thank you for the very good comment. Please refer to page 2 line 31-32.
- Intro. L43. FDA in USA is well known, but what does TGA in Australia mean? Thank you for the great comment. Please refer to page 3 line46-48.
- M & M. para. 2.1. How were the HBsAg carriers identified before they entered the PROLIFICA study? POC test described already. Thank you for the great comment, it is much appreciated. Please see page 3 line 3-16.
- Did they have symptoms of liver disease, were they blood donors or contact persons to HBV carriers? Thank you for your question it is highly appreciated. The study participants are majorly HBV chronic carriers from blood donors and the general community with no liver disease symptoms. Please see page 4 line 3-18.
- How sensitive were the tests for identification and during study, i.e., the limit of detection for HBsAg? Thank you for the beautiful comment. Please see the diagnostic performance, sensitivity and specificity of the POC on page 4 line 8-10.
- Was HBsAg quantified? Study population. Thank you for the great comment. The HBsAg was not quantified a POC test was used. Please see page
- Were HBeAg or HBcrAg results available? Thank you for the question. These results were not available.
- M & M. L41, 47 and later. The Cobas TaqMan test for HBV DNA is from Roche, not from Abbott. Thank you very much for highlighting this. All were amended accordingly and highlighted in blues through the text.
- Results. L5-10. This para. 2.1 is not well explained. It remains open which aliquot of plasma in the DBS samples with 3, 4, and 6 punches ended up in the Xpert test and how much plasma corresponded to amount of HBV DNA given into the Xpert test. Are the differences between DBS and plasma in the quantification only due to the different dilutions or different sizes of aliquots or is HBV DNA lost during elution of the DBS? Thank you for the beautiful comment. Please see page 4 line 4 to 8.
- Results L11-20. The text should refer to the two following points or the paper should be corrected correspondingly. Thank you for your suggestion, we tried to change it as you recommended. Please see page 4 line 4-18.
- Which kind of samples were tested? Were these all HBV DNA positive in previous highly sensitive tests? Yes all HBV DNA samples were initially tested with a very sensitive test as described previously.
- Table 2. Why are 58 plasma samples in the table but only 56 for the DBS samples? If 2 results are missing in the DBS group, these two corresponding plasma samples should have been deleted. Thank you for the great observation. Two DBS samples were excluded during the analysis since they could not be retested due to insufficient sample volume.
- Fig. 1B. Why does in this figure of the Bland-Altman plot the X-axis show the mean difference of plasma minus DBS viral loads whereas in all four other plots the mean value of the two variables is shown? Thank you for the beautiful comment. The error was corrected, refer to figure 1B.
- Results L22-78.
- Table 4 and L26, 46, 47. Was the Cobas Taqman or the Abbott test used for HBV DNA? Cobas Abbott does not exist. The figure legends show “Abbott” and the discussion mentions in L61 also Abbott, but is this true? Thank you for the beautiful comment. We Yes the COBAS TaqMan was used for HBV DNA. We have changed all the errors pertaining to the errors you rightly sighted throughout the text.
- L36. Table 3 should be corrected to table 4. The large differences in the proportion of positive results between the three tests would deserve some explanatory sentences in the main text. Thank you for the brilliant suggestion. Please refer to page 7-8 line 36-44
- L40-44. A brief explanation on what the “bias” practically means would be helpful. Does it mean that the in-house test measures on the average DNA levels by 1.173 log IU/mL lower than the Xpert test? Thank you for the great comment. Your suggested has been amended. Please refer to page 8 line 48-50.
- Fig.3 A & B. The letters at the Y-axis not readable and one of the two texts is probably wrong in both panels. Thank you for your comment. Please see figure 3A and B.
- L 49. It is interesting that the Xpert test yields obviously slightly higher values (mean bias = 0.316) than the Abbott test (if I guess right from the confusing graphs) but 5 samples positive in the Abbott test were not detected with Xpert test. Yes it is known that in between 2 excellent commercial tests the discrepancies remains, this is very frequent unfortunately.
- L57-65. It must be cleared whether the commercial test was from Roche or Abbott. Thank you for your beautiful comment. We have made the change of brand to Roches.
- Fig. 4B and L61-65. The figure shows clearly that the commercial test yielded in the average 1.421 log IU/mL higher values than the in-house PCR. However, the bias is positive with 1.421. Does this also mean that the in-house test was considered to be the reference test which would be not appropriate? Thank you for your question. Yes the in-house is the gold standard.
- Results para. 3.8
- L84. What happened with the 51 other patients with >200,000 IU/mL who were not eligible within 6 days? Did they receive a therapy later and how much later? Thank you for the question. Your question was addressed in page 9 line 97.
- L97. How does the 55% rate per week fit to the 1 in 5 lost cartridges? Thank you for the keen observation. The error was amended in page 10 line 109.
- Discussion
- L11. Shouldn’t the bias be -1.421? See point 10g. Thank you for your beautiful observation. Please see page 11 line 10.
- L14. Is a bias of 1.831 small? Plasma gives 68 times higher IU/mL values than the DBS. This could decide whether therapy is planned or not. The bias is not small but after calculation we should not miss the patients in need of care as compared to the international recommendations.
- L29. Do you mean with “assays” the commercial assays? Thank you for your question. We meant the two assays that were used. Please refer to page 11 line 29-30.
- L30. A bias of -1.831 is not excellent concordance, because the HBV DNA level decides about therapy. Yes indeed, but after calculation we should not miss the patients in need of care as compared to the international recommendations.
- L82, 94, and 102. See above under 12 b). The DBS sample are inferior not only in sensitivity, but also for exact quantification, and that should be acknowledged. At least, the limitations should be discussed in more detail. The DBS is just a surrogate to serum/plasma samples for remote areas and even if not perfect they are a wat to reach a number of patients too far from clinical centers.
Comments on the Quality of English Language
The language is often insufficient and partially not understandable.
Submission Date
09 September 2024
Date of this review
20 Sep 2024 22:43:47
Reviewer 3 Report
Comments and Suggestions for Authors
The manuscript improved.
My additional critical comments:
- Is this methodology popular all over the world?
- Please compare your results with those from other countries, if available.
- Please check the grammatical and typographical errors.
Author Response
Reviewer 3
The manuscript improved.
My additional critical comments:
- Is this methodology popular all over the world? GeneXpert is expanding especially after COVID pandemic, but know appear as useful for HIV, tuberculosis, it is present everywhere but especially adapted for low/middle income countries with limited access to clinical settings.
- Please compare your results with those from other countries, if available. Sorry but is too early for HBV since this study is a pioneer, new ones are just starting.
- Please check the grammatical and typographical errors. This was done, sorry about that.
Round 2
Reviewer 1 Report
Comments and Suggestions for Authors
Thanks to the authors for revising the manuscript. The manuscript was improved.
Minor comment
In Table 7, please correct “(IQR (12.5- 8)” to “(IQR 8-12.5)”, “(IQR 113-179” to “(IQR 113-179)”, “(122- 28)” to “(IQR 28-122)”, “(IQR 6.5-5.5)” to “(IQR 5.5-6.5)”, and “(16-37)” to (IQR 16-37)”.
Author Response
For research article
|
Response to Reviewer 1 Comments
|
|||||||||||||
|
|
|
||||||||||||
|
3. Point-by-point response to Comments and Suggestions for Authors |
|||||||||||||
|
Comments 1: Minor comment In Table 7, please correct “(IQR (12.5- 8)” to “(IQR 8-12.5)”, “(IQR 113-179” to “(IQR 113-179)”, “(122- 28)” to “(IQR 28-122)”, “(IQR 6.5-5.5)” to “(IQR 5.5-6.5)”, and “(16-37)” to (IQR 16-37)”.
|
|||||||||||||
|
Response 1: Thank you for pointing this out. We agree with this comment. Therefore, we have amended the interchanges numbers in the table as recommended and highlight them in yellow in table 7.
|
|||||||||||||
For review article
|
Response to Reviewer X Comments
|
||
|
1. Summary |
|
|
|
Thank you very much for taking the time to review this manuscript. Please find the detailed responses below and the corresponding revisions/corrections highlighted/in track changes in the re-submitted files. [This is only a recommended summary. Please feel free to adjust it. We do suggest maintaining a neutral tone and thanking the reviewers for their contribution although the comments may be negative or off-target. If you disagree with the reviewer's comments please include any concerns you may have in the letter to the Academic Editor.]
|
||
|
2. Questions for General Evaluation |
Reviewer’s Evaluation |
Response and Revisions |
|
Is the work a significant contribution to the field? |
|
[Please give your response if necessary. Or you can also give your corresponding response in the point-by-point response letter. The same as below] |
|
Is the work well organized and comprehensively described? |
|
|
|
Is the work scientifically sound and not misleading? |
|
|
|
Are there appropriate and adequate references to related and previous work? |
|
|
|
Is the English used correct and readable? |
|
|
|
3. Point-by-point response to Comments and Suggestions for Authors |
|
|
|
Comments 1: [Paste the full reviewer comment here.]
|
||
|
Response 1: [Type your response here and mark your revisions in red] Thank you for pointing this out. I/We agree with this comment. Therefore, I/we have….[Explain what change you have made. Mention exactly where in the revised manuscript this change can be found – page number, paragraph, and line.] “[updated text in the manuscript if necessary]” |
||
|
Comments 2: [Paste the full reviewer comment here.] |
||
|
Response 2: Agree. I/We have, accordingly, done/revised/changed/modified…..to emphasize this point. Discuss the changes made, providing the necessary explanation/clarification. Mention exactly where in the revised manuscript this change can be found – page number, paragraph, and line.] “[updated text in the manuscript if necessary]” |
||
|
4. Response to Comments on the Quality of English Language |
||
|
Point 1: |
||
|
Response 1: (in red) |
||
|
5. Additional clarifications |
||
|
[Here, mention any other clarifications you would like to provide to the journal editor/reviewer.] |
||
Reviewer 2 Report
Comments and Suggestions for Authors
The authors have responded to all my comments adequately and have corrected the paper thoroughly. Twenty of the 26 specific points have been corrected. In three case, no sufficient data were available. In three further points the authors did not agree with my opinion which is absolutely accetable.
Author Response
For research article
|
Response to Reviewer 2 Comments
|
||
|
1. Summary |
|
|
|
Thank you very much for taking the time to review this manuscript. |
||
|
No comment was given. |
||
|
We thank the reviewer very much for taking the time to review this manuscript, we appreciate his/her time. |
